# A Review of Low-Frequency Active Vibration Control of Seat Suspension Systems

**Yuli Zhao and Xu Wang *** 

School of Engineering, RMIT University, Bundoora, VIC 3083, Australia

* Correspondence: xu.wang@rmit.edu.au

**Abstract:** As a major device for reducing vibration and protecting passengers, the low-frequency vibration control performance of commercial vehicle seating systems has become an attractive research topic in recent years. This article reviews the recent developments in active seat suspensions for vehicles. The features of active seat suspension actuators and the related control algorithms are described and discussed in detail. In addition, the vibration control and reduction performance of active seat suspension systems are also reviewed. The article also discusses the prospects of the application of machine learning, including artificial neural network (ANN) control algorithms, in the development of active seat suspension systems for vibration control.

**Keywords:** vibration; active control; low frequency; suspension; seating system; machine learning

## 1. Introduction

Modern research has shown the potential hazards of whole-body vibration (WBV). Based on previous studies, Krajnak (2018) [1] summarized the various diseases that may be caused by whole-body vibration, including back and neck pain, neuropathy, cardiovascular disease, digestion disorders, and even cancer.

As a risky occupation, the drivers of heavy commercial vehicles are prone to prolonged exposure to low-frequency whole-body vibrations generated from road excitation, which could influence drivers' comfort or even adversely affect their health. According to [2], the long-term operation of heavy commercial vehicles with low-frequency vibration can cause diseases of the muscles, bones, digestive system, and even the visual system. This is because low-frequency vibrations might cause resonance of the organs and tissues in the human body, and this kind of vibration energy must also be absorbed and dissipated by the body. According to [3], due to the high social cost of musculoskeletal diseases caused by working environments with low-frequency vibrations, Europe has issued regulations requiring that the vibration level of a working vehicle must be evaluated to provide a healthy and safe operation environment, where the max accelerations of 0.5 and 1.5 m/s$^2$ are, respectively, set for 8 hours action and limit values. In [4], it was proved that the WBV affects the posture control of the human body and may cause health risks to the muscular system and spine. In [5], it was shown that if the exposure to low-frequency vibrations produced by commercial vehicles were more than 8 hours every day, conventional vehicle seating with passive suspension could not protect the driver's body from the effects of WBVs. According to a medical research report by Johanning (2011) [6], low-frequency vibrations may cause resonances of human organs and tissues. There, it was claimed that back pain disease is one of the most common occupational injuries because the lower back of the human body is sensitive to low-frequency vibrations of 4–10 Hz. Therefore, long-term exposure to large amplitude low-frequency vibrations can cause back pain disorders, especially common diseases of the lumbar joints. These diseases include degenerative spinal changes, lumbar disc herniation, and sciatic nerve injuries, and according to ongoing medical reports, these diseases are common among tractor drivers, truck drivers, bus drivers, and other commercial heavy machinery operators who are often exposed to vibrations throughout the body.

The diagram (Figure 1) shows the vibration amplitude of commonly used heavy commercial machinery. According to the international standard ISO 2631-1 1997, as the vertical acceleration increases, the ride comfort decreases and the WBV level increases, which also increases health risks. Therefore, the development of an efficient vibration reduction seating system is a practical and effective way to protect drivers. Researchers have studied the vibration control of vehicle seating systems as well as the various effects that vibrations and vibration transmission have on the human body. In [7], the seats of 100 commercial heavy-duty vehicles from 14 different categories were tested and evaluated for vibration isolation performance. Two groups of Seat Effective Amplitude Transmissibility (SEAT) values calculated by the weighting parameters of different standards (BS6841 and ISO2631) showed that the average SEAT values of these seats were all less than 100%, indicating that they provided a certain degree of protection. This article also proposed improving the dynamic performance of seats, which can reduce the severity of WBV exposure in many working environments. In addition to the dynamic characteristics of seats, the dynamic response of the human body under vibration excitation is also a topic of considerable interest. In [8], 41 male and 39 female subjects between 18 and 65 years of age were selected to participate in an experiment to study the factors that may affect the apparent mass of the human body, which is a method that can be used to present comfort levels and to estimate WBV levels. According to this study, aging can affect the resonant frequency of the human body and the transmission ratio of vibration. Further, gender and body mass index (BMI) are also factors that affect vibration transmissibility. In addition to research on the whole seat, the seat cushion, as an important vibration isolation device, was examined in [9]. There, they investigated the seat cushion–body interaction by measuring and analyzing the contact force distribution and the contact area between the human body and the seat cushion when vibration is experienced. It was found that the pressure distribution at the interface between the body and the cushion showed strong asymmetry in terms of dynamic contact force, and the effective contact area was affected by the nonlinear characteristics of the cushion itself and the characteristics of the soft tissue of the human body. Further, under large vibration excitation, a seat cushion with high stiffness, a large damping coefficient, and large static deflection is able to effectively reduce the transmission of vibration.

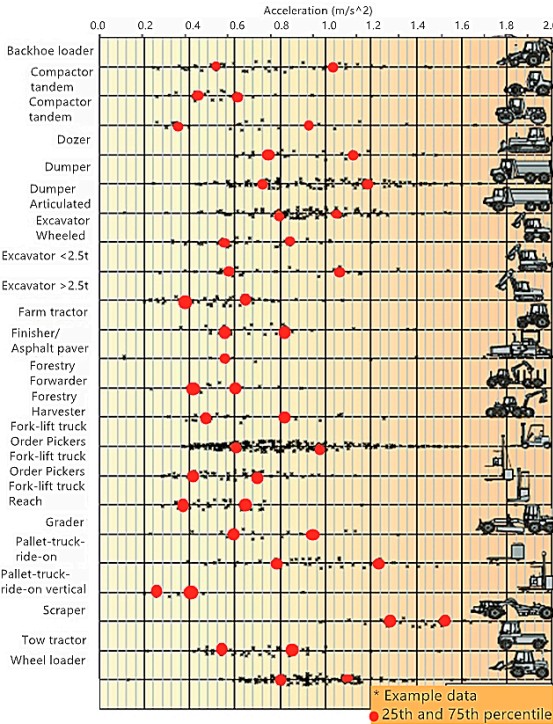

**Figure 1.** Vibration measurement examples of on- and off-road vehicles [6].

In terms of seat vibration isolation measurements, Mozaffarin et al. (2008) [10] designed an active dummy technique (Figure 2), which adopted lateral and longitudinal actuators to produce forces in the vertical and lateral directions, respectively, to simulate the dynamic response of three different human body masses by reproducing the equivalent dynamic mass.

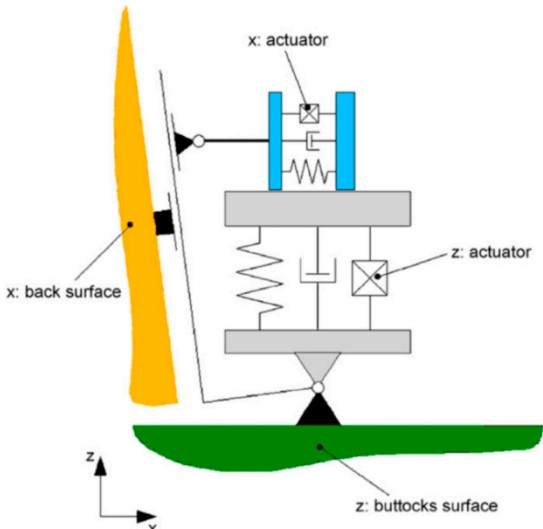

**Figure 2.** The schematic diagram of MEMOSIK V [10].

Of course, in all of these works, determining how to control vibration is the most critical aspect. Three current mainstream research directions in this area are passive, semiactive, and active vibration controls.

Passive vibration control of a seating system can reduce vibration by using conventional spring and damper components in the seat suspension system, but due to its characteristic limitations, vibration control that targets multiple frequencies cannot be achieved even with a well-tuned traditional positive spring–damper system. Therefore, a quasi-zero static stiffness seat suspension system based on the combination of negative stiffness and positive stiffness springs was proposed by Le and Ahn (2013) [11] to improve the vibration control efficiency of a seating system, since the characteristics of high static stiffness and low dynamic stiffness can be used to eliminate seat vibration. In [12], a poly-optimal solution was sought for a seating system combined with a pneumatic spring and damper. This improved the performance of the traditional passive seating system by attenuating low-frequency vibrations in the frequency range of 0–4 Hz.

Semiactive vibration control of a seating system utilizes the characteristics of magnetorheological [13–16] and the electrorheological materials [17], which can change the stiffness or Young's modulus under magnetic field variations and achieve vibration control in a specific frequency range. In [18], a negative stiffness seat suspension system combined with a pneumatic spring and stiffness control mechanism was proposed, and the related control algorithm that affects device stiffness variations associated with position and velocity data evaluation was designed. The semiactive vibration control method can achieve vibration control in a relatively certain frequency bandwidth with less energy consumption and fewer costs than the other two methods.

In [19], the difference between seating systems with active electromagnetic seat suspension and passive seat air suspension in reducing the WBV level and improving comfort was compared. The experimental results showed that the active electromagnetic suspension performed better at vibration reduction than the passive air spring suspension. In particular, passive suspensions may increase the amplitude of lateral vibration, which also has a negative effect on driver comfort.

In terms of structure, in addition to traditional shock absorbers, semiactive seat suspensions have also been designed in different styles. Bai et al. (2017) [14] designed an integrated semiactive seat

suspension that included a swing mechanism (Figure 3) that converts longitudinal and vertical motion into rotational motion and a torque-controlled rotary magnetorheological (MR) damper operated in a pure shear mode to attenuate vertical and longitudinal vibrations. Ning et al. (2019) [20] proposed a new semiactive seat suspension based on the variable admittance (VA) concept and designed a rotating VA device based on the MR damper. A random vibration test showed that the semiactive seat suspension had excellent low-frequency vibration cancellation performance. The frequency-weighted root-mean-square (FW-RMS) acceleration of the seat was reduced by 43.6%, indicating that ride comfort was greatly improved. Ning et al. (2018) [21] developed a semiactive vibration control seating system based on an energy harvest device with variable external resistance. The energy regeneration seat device included a three-phase generator and a gear reducer mounted at the centre of the scissor-like structure of the seat, and the vibrational energy was collected directly from the rotational motion of the scissor-like structure. An H-infinity-state feedback controller was designed for a semiactive vibration control seating system, and the FW-RMS acceleration was reduced by 22.84% compared with passive vehicle suspension. At the same time, the generated RMS power was 1.21 W.

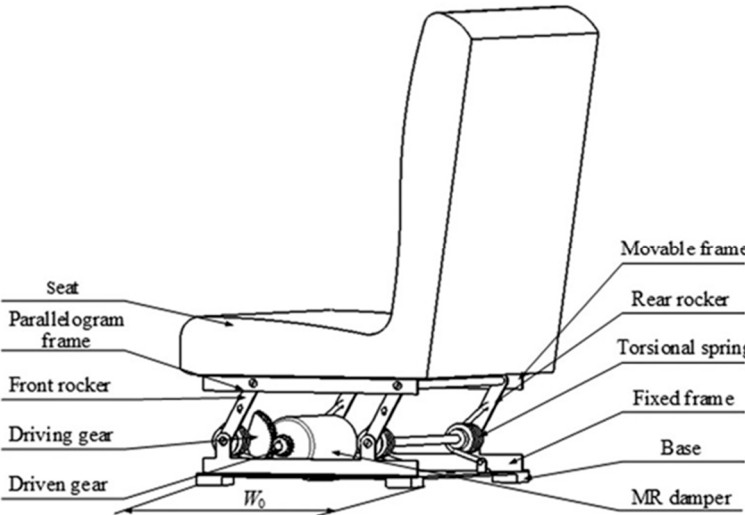

**Figure 3.** Semiactive seating system vibration controls with a magnetorheological (MR) damper [14].

For the controller design, Sky-hook control theory [22–25], H-infinity control algorithm [26–32], nonresonance theory [15], on-off control strategy [33], fuzzy control theory [34–36], optimal control theory [37–39], lyapunov control scheme [40], PID control algorithm [41] and sliding-mode control algorithm [42] are applied for the design of the controllers.

Compared with controller designs based on a single traditional adaptive control algorithm, an increasing number of controllers integrating multiple control algorithms with semiactive vibration control seating systems have been proposed to improve vibration attenuation performance. Phu et al. (2015) [43] developed a new adaptive fuzzy controller combining the H-infinity and sliding-mode control algorithms for a semiactive seat suspension with an MR fluid damper. This controller features a fuzzy control method that does not require an accurate dynamic model, even in a dynamic system with an uncertain environment. Phu et al. (2017) [44] designed a new adaptive hybrid controller integrating the H-infinity control algorithm, the sliding-mode control algorithm, and the Proportional-Intergral-Derivative (PID) control algorithm with the vibration attenuation of a semiactive seating system. This controller features a combination of the Hurwitz constant matrix as components of the sliding surface and the H-infinity algorithm with robust stability. In addition, a fuzzy logic module based on the interval type-2 fuzzy logic system was established, and a model was characterized by on-line clustering considering external interference. Phu et al. (2017) [45] proposed a new hybrid controller combining a neural fuzzy control module, a Proportional-Integral (PI) control module, and a sliding mode control module to control a semiactive seat suspension with an MR damper. The interval

type-2 fuzzy model with an on-line rule updating function was adopted, and a granular clustering method was used to find data for the initial fuzzy set used to support the fuzzy model. Compared with conventional controllers, the proposed controller can provide better stability for vibration control performance. Nguyen et al. (2015) [46] designed a novel neuro-fuzzy controller (NFC) for a semiactive seat suspension system with an MR damper. This adaptive neuro-fuzzy inference system (ANFIS) is based on an algorithm called B-ANFIS, which is combined with a fuzzy inference system (FIS). Compared with the skyhook control theory, the NFC is better at improving the ride comfort of the vehicle. In addition, the NFC's ability to track trajectories and transient response characteristics is superior to that of conventional skyhook controllers. Phu et al. (2017) [47] proposed a new adaptive fuzzy controller based on inversely fuzzified values related to the H-infinity control algorithm to control the vibration of a semiactive seat suspension system with an MR damper and a Riccati-like equation with fuzzified values to enhance system robustness.

This paper aims to review and classify the active vibration control of seating systems proposed in previous works that have active seat suspension. In addition, the algorithms and structures of the active controllers are analyzed and discussed. Finally, the development of artificial neural network (ANN) controllers and their applications in active vibration control in seating systems are also discussed.

## 2. Seating Systems with Active Suspension

### 2.1. Experiments with Prototypes

Active seat suspension prototypes in several research projects have used electromagnetic, hydraulic, or air actuators to generate a corresponding compensation force for vibration cancellation in a finite number of frequencies, thereby reducing the vibration acceleration amplitude and improving the comfort of the seating system. According to [48], using an active vibration control suspension in a vehicle has better vibration cancellation performance than passive seat suspension. In [49], comparative experiments demonstrated that active and semiactive seat suspensions could improve comfort by approximately 50% compared with passive seat suspensions.

The actuators used for active seat suspensions generally fall into three categories: electromagnetic actuators using linear or conventional rotating motors, hydraulic actuators using hydraulic servos, and air actuators using air springs. Among these three types of actuators, electromagnetic actuators have attracted the most attention because they have good dynamic responses; do not require an additional hydraulic servo system, tubes, or compressors; and have small space requirements. The large output of hydraulic actuators makes them easy to carry with greater mass. The air actuator has a good vibration isolation effect in high frequencies.

#### 2.1.1. Pneumatic Actuator

Stein (1997) [50] developed an active seat suspension using a pneumatic spring and a corresponding feedback controller. According to the results, the active seat suspension using vibration compensation can reduce the vibration amplitude by 10 dB, which is about 3 times. The vibration transmission rate is reduced by 30%–40% compared with a conventional passive seat suspension.

In another research project, Stein (1997) [51] proposed an active vibration control seating system to attenuate low-frequency vibration with an active seat suspension (Figure 4) composed of a pneumatic spring and a related linear control system. The conventional metal spring in the system is used to carry the static load, and the pneumatic spring is used to generate damping and compensating forces to mitigate vibration. The pneumatic spring is driven by compressed air which is controlled through a proportional valve. The benefit of such a parallel structure is that the energy consumption of the whole system can be reduced. Two acceleration sensors and one displacement sensor were used in the experimental equipment, and the electric signals generated by the excitation were respectively collected by the corresponding controller and summed at the sum point to control the actuator. There is no actual damper in this system, and the function of the damper is replaced by absolute damping,

which is generated from the air spring through the skyhook control method. This active suspension needs to consider the complex state and parameters of the entire pneumatic subsystem and take into account the state equation of compressed air. In addition, the flow rate of the air, the corresponding pressure changes in the air spring and the corresponding forces are also considered. For the control system, a simple and practical linear controller system is used that is based on feed-forward and feed-back algorithms to reduce the vibration effect on the human body by controlling the static position, damping force, and compensation force of the seating system. Researchers have found that this active pneumatic spring seat suspension can reduce vibration transmissibility by approximately 8–10 dB with the feed-forward compensation path for a mass of 80 kg. In an experiment, by adjusting the parameters of the pneumatic spring, it was found that the active vibration control of the seating system performed well in a frequency range less than 4 Hz. However, this control system needs to work under an ideal condition.

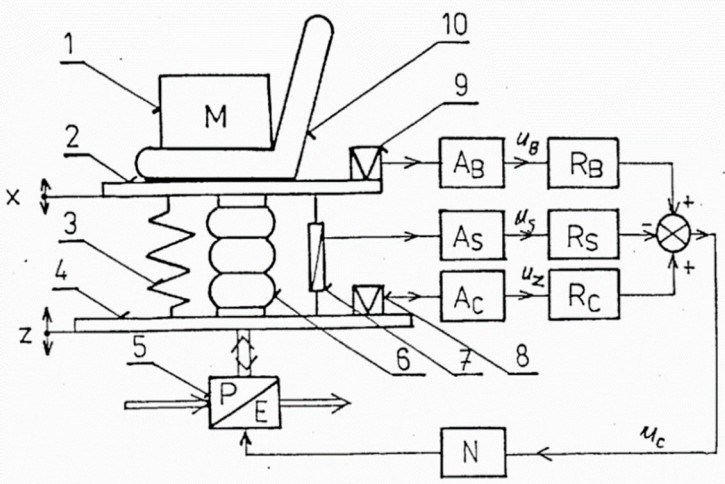

**Figure 4.** Active seating system vibration controls with a parallel spring structure [51].

An active seat suspension incorporating a hydraulic shock absorber and an active pneumatic spring (Figure 5) was developed in [52–54]. In this seating system, a hydraulic shock absorber is connected to a scissor-like frame for vibrational energy absorption, while a pneumatic spring is attached to the bottom of the seat and a rod of the frame to produce the corresponding compensation force. The air spring is inflated and deflated by using compressed air and a proportional air valve. The advantage of this configuration is that configurations of commonly existing vehicle seating systems can be used without large modifications. Based on this type active seat suspension system, a robust controller (Figure 6) that can work with different mass loads was proposed by Maciejewski et al. (2010) [52] for an active seat suspension. In the design of the controller, the author used a triple feedback loop system to detect the acceleration, the relative speed, and the displacement of the suspension system and to control the system. The results of the experiment demonstrated that active vibration control of this seating system could be used to reduce the amplitude of the vibration by half compared with conventional passive seating systems at a resonant frequency. The system can control the suspension well in the range of 0.5–4 Hz and can reduce response amplitudes under different mass load conditions.

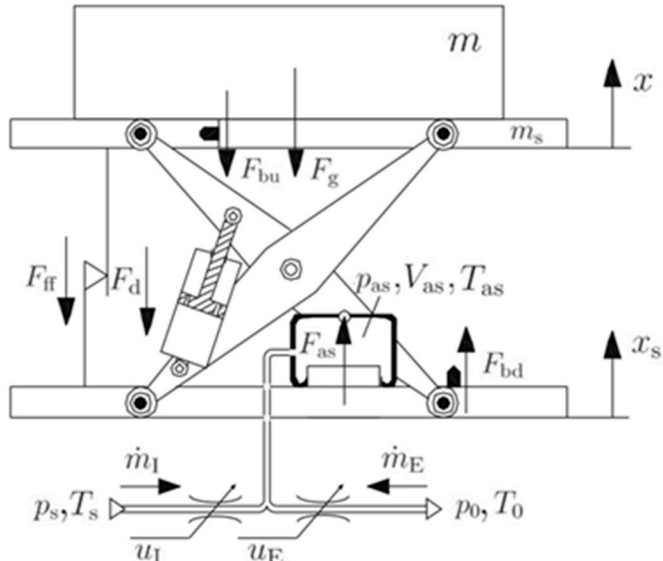

**Figure 5.** A seating system with an active pneumatic spring suspension [52].

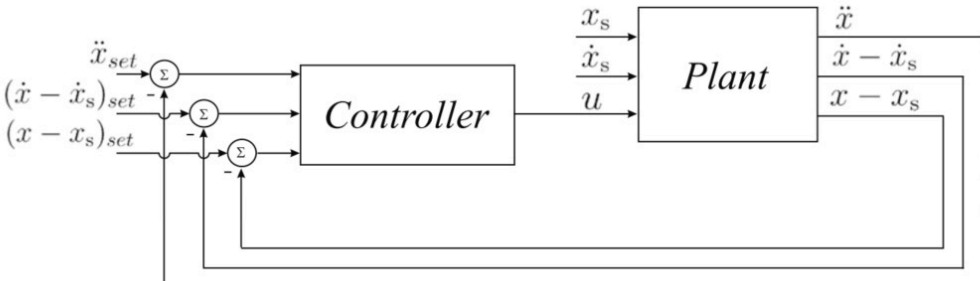

**Figure 6.** Schematic of the triple feedback loop controller [52].

With the same active seat suspension, Maciejewski et al. (2014) [54] proposed an adaptive controller for an active seat suspension (Figure 7). This is a multicontroller approach to control the entire system, where the primary controller is used to calculate the force required to reduce the vibration. The inverse model is used to calculate the effective area of the proportional control valve. The application of the inverse model in the controller can directly derive the input signal according to the required force. The Proportional-Derivative (PD) predictor is used to generate the corresponding control signal to speed up the controller. Finally, the adaptive mechanism is able to estimate the load mass based on the inflation and deflation of the pneumatic spring. This active seat suspension is able to achieve good vibration control performance compared with a passive system, having a load range from 50 to 150 kg at a resonant frequency of 1.3 Hz. The advantage of this controller is that the adaptive control itself makes the system quickly return to stability by estimating the initial suspended load. The shortcoming of the controller is that the complexity of the multicontroller may delay signals, and in the case of high road roughness, the vibration control performance of the seating system is degraded.

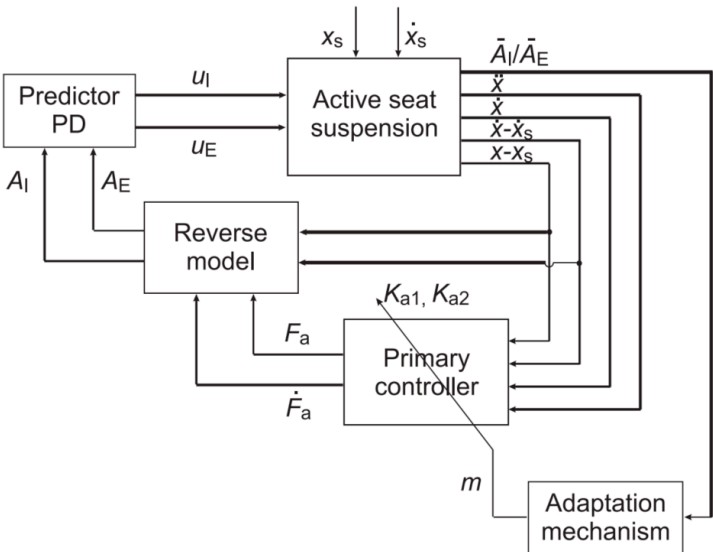

**Figure 7.** Schematic of the multicontroller [54].

Maciejewski et al. (2018) [55] developed a horizontal active seat suspension using pneumatic muscles (Figure 8) for horizontal vibration control. This original control system (Figure 9) combined a primary controller and an inverse model module to provide a control signal to the pneumatic muscles, and a PD control module was used to speed up the signal. According to the final results, the proposed active seat suspension performed better than a passive seat suspension for vibration attenuation in the 1–10 Hz frequency range.

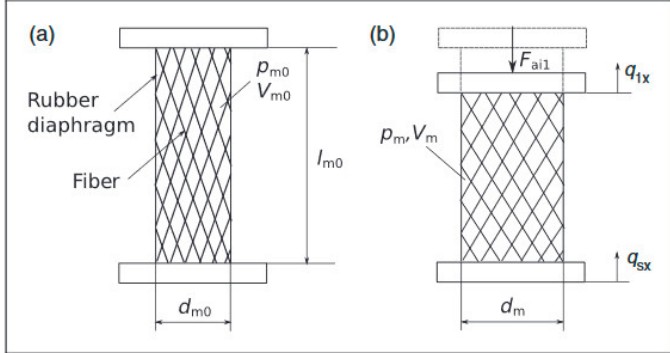

**Figure 8.** (**a**) The pneumatic muscle at the nominal length and (**b**) After contraction [55].

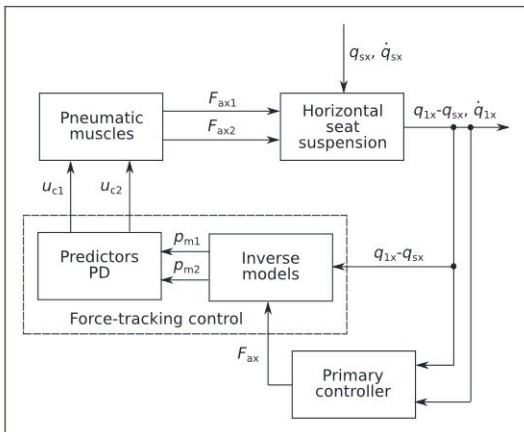

**Figure 9.** Block diagram of the control structure of the vibration control system [55].

The utilization of pneumatic springs as actuators has some advantages, such as being simple, reliable, and compact. At the same time, their low response speed, poor control precision, and dependence on a compressor pipeline hinder their actual use.

### 2.1.2. Hydraulic Actuator

Stein and Ballo (1991) [56] developed an active seat suspension (Figure 10) using a hydraulic actuator. In this system, the hydraulic actuator is controlled by a solenoid valve to control the direction of the force generated by the actuator. The control system generates a corresponding compensation force to reduce the vibration based on signals collected by two acceleration sensors and a displacement sensor. A PI controller was designed for the active seat suspension. Since the acceleration sensor and amplifier are integrated for the displacement of the positioning system, the system can be simplified using only one accelerometer placed on the chassis. This structure greatly simplifies the control system. This structure can also be achieved by connecting two first-order low-pass filters in series or by using a second-order analog circuit to adopt a very low resonant frequency (Figure 11).

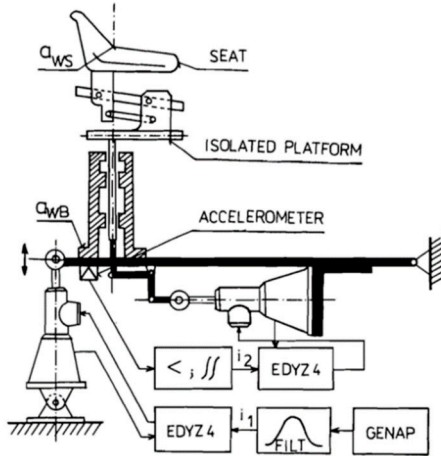

**Figure 10.** Active hydraulic control of a seat suspension system [56].

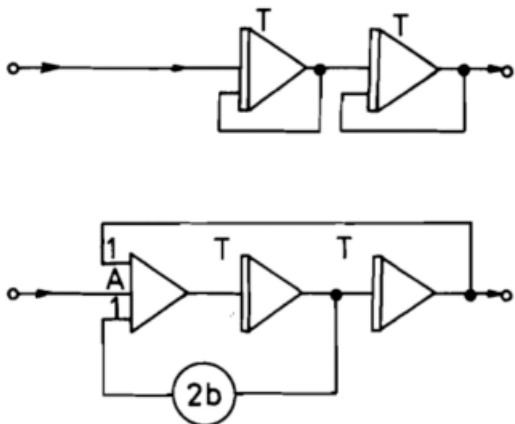

**Figure 11.** Schematic of the control system [56].

In this study, the delay of hydraulic performance was also considered and solved by using the cut-off frequency method. In addition, the hysteresis effect of the hydrodynamic device was also carefully considered in the process of controller design. Depending on the result, the active seat suspension with the controller can reduce the acceleration Power Spectral Density (PSD) amplitude up to 16 dB at a frequency of 2 Hz compared with a passive system.

Active hydraulic control has the advantages of a large output force and high control precision, but the huge hydraulic pipeline and servo system of hydraulic control also greatly limit its application.

2.1.3. Electromagnetic Actuator

A new active seating vibration reduction structure (Figure 12) was proposed in [57–60]. This seating system is based on the common scissor-like structure of commercial vehicle seating systems, using an inexpensive conventional rotary electric motor instead of an expensive linear motor as the actuator. The 1:40 gearbox allows the 400 W Panasonic DC motor to produce a torque output of 52 Nm. Moreover, due to the enlarged gearbox, the internal friction of the active suspension is greater than that of a conventional suspension system. The system can save some space because there is no need to install a conventional shock absorber. In addition, the spring stiffness of the seat is carefully selected to keep the resonant frequency of whole seating system lower than 4 Hz, which is the most sensitive vibration frequency range of human body and may cause an uncomfortable feeling.

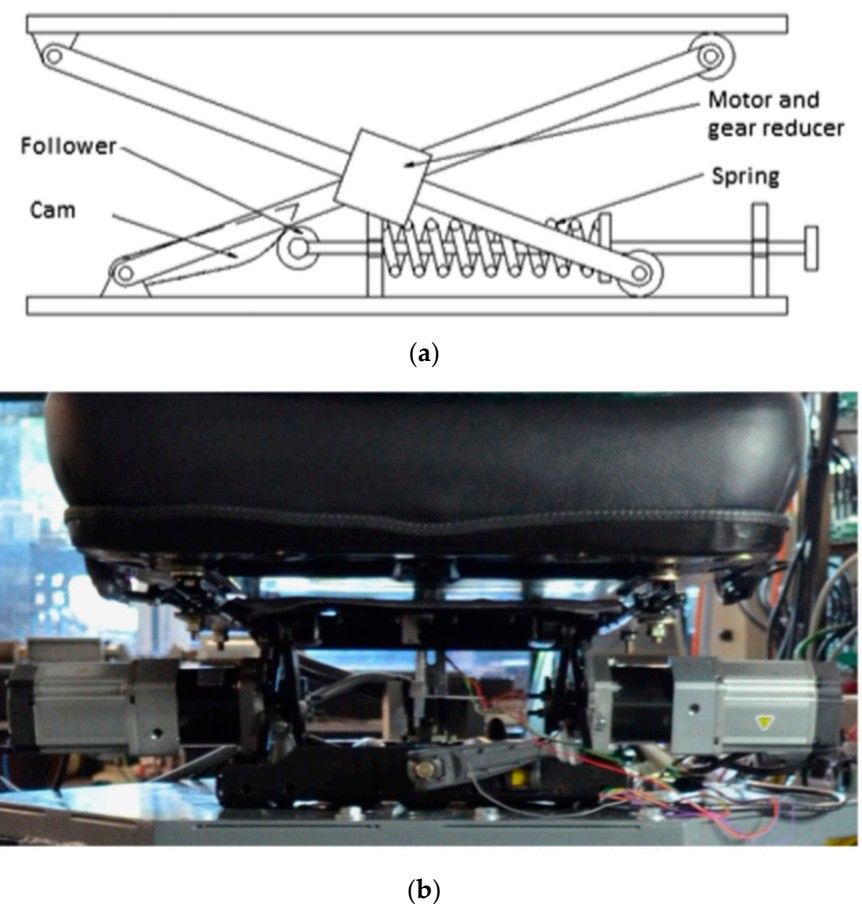

(a)

(b)

**Figure 12.** (**a**) The schematic of the seat structure (**b**) Front view of the seat structure [58].

In terms of the control system, an H-infinity algorithm based on output feedback was developed by Ning et al. (2016) [57] to reduce the seating system vibration. The feature of this project was the estimation of the friction generated by the active vibration control system. A special controller with friction compensation was employed to improve the precision of the control, which can affect the performance of the vibration control. The final test results showed that in the low-frequency range, from 3 to 5 Hz, the driver's body RMS acceleration can be reduced by more than 35%.

In another study, Ning et al. (2016) [58] proposed an H-infinity controller with friction compensation to control the previously mentioned active seat suspension. Due to the usage of the friction observer based on the acceleration measurement, the H-infinity controller can be more

sensitive to the response of vibration excitation and can also improve the vibration control performance. The results of the experiment showed that the entire vibration control system could reduce vibration very well in a frequency range from 1 to 4.5 Hz. According to the ISO 2631 standard, the FW-RMS value of the vibration is reduced up to 35.5% by the vibration control system compared with a conventional well-tuned passive suspension system. The RMS energy consumption of 3.82 Watts indicates that the system's energy usage is very low.

In [59], a terminal sliding mode controller based on a space observer and a disturbance observer was proposed by Ning et al. (2017) to control an active seat suspension system (Figure 13). In the case where the suspension acceleration and relative displacement are measurable, but the absolute seat speed is immeasurable, the settings of the disturbance and state observers can reduce the switching gain of the controller. The controller proposed in this paper had better control performance than the state feedback terminal sliding mode controller and could improve the comfort of the seating system. According to the experimental results, FW-RMS and Vibration Dose Value (VDV) were reduced by 34% and 33%, respectively.

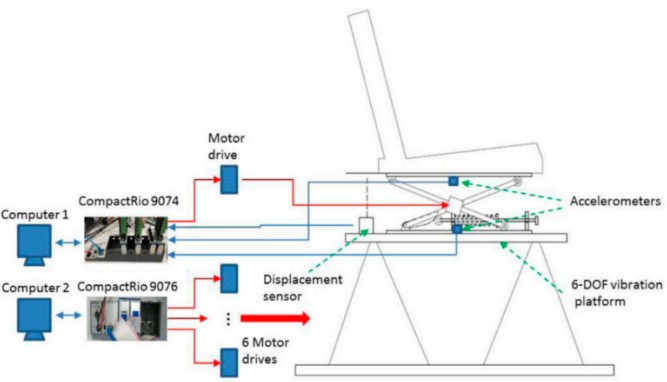

**Figure 13.** The experimental setup of the system with the terminal sliding mode controller [59].

With the same active seat suspension system, a disturbance observer based on the Takagi–Sugeno (TS) fuzzy controller was proposed by Ning et al. (2017) [60] for active seat suspension vibration control (Figure 14). The controller used a closed-loop feedback control with acceleration and seat suspension displacement measurement signals to achieve good adaptability and robustness. The disturbance observer can estimate disturbances caused by friction and model simplification. The TS fuzzy control improves the vibration reduction performance of the controller by estimating load changes. During the experiment, the controller worked well in vibration frequencies below 4 Hz. Two different loads of 55 and 70 kg could achieve a good response through the controller. The active seat suspension control was able to reduce the RMS acceleration by more than 45% compared to a well-tuned passive seat suspension.

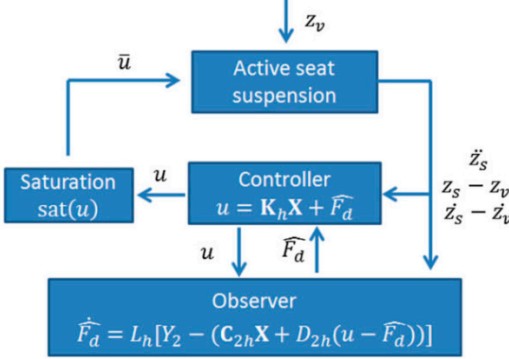

**Figure 14.** Block diagram of the Takagi–Sugeno (TS) controller with a disturbance observer [60].

Based on previous research, Ning et al. (2018) [61] proposed active vibration control of a multi-degree-of-freedom (multi-DOF) seating system with a double-layer structure (Figure 15) and an associated controller. A second layer was added to the preceding scissor-like structure to control the vibration of pitching and rocking. The second layer consists of a universal joint and four support springs. The universal joint connects two conventional rotating motors and a gearbox to generate a torque of 52 Nm to remove the rolling and pitching vibrations. The spring system is responsible for supporting the static load. This is a simple and efficient active vibration control of a multi-DOF seating system compared with other active vibration controls of multi-DOF seating systems. For the control system, a sliding mode controller to reduce the roll vibration was designed. This algorithm has the advantages of fast convergence and good robustness and can be used to control the vibration in the swinging direction. In the design, for the swaying and pitch, the controller uses the minimum lateral acceleration and the minimum rolling angle as the control target, while the vertical vibration control uses the vertical acceleration as the control target.

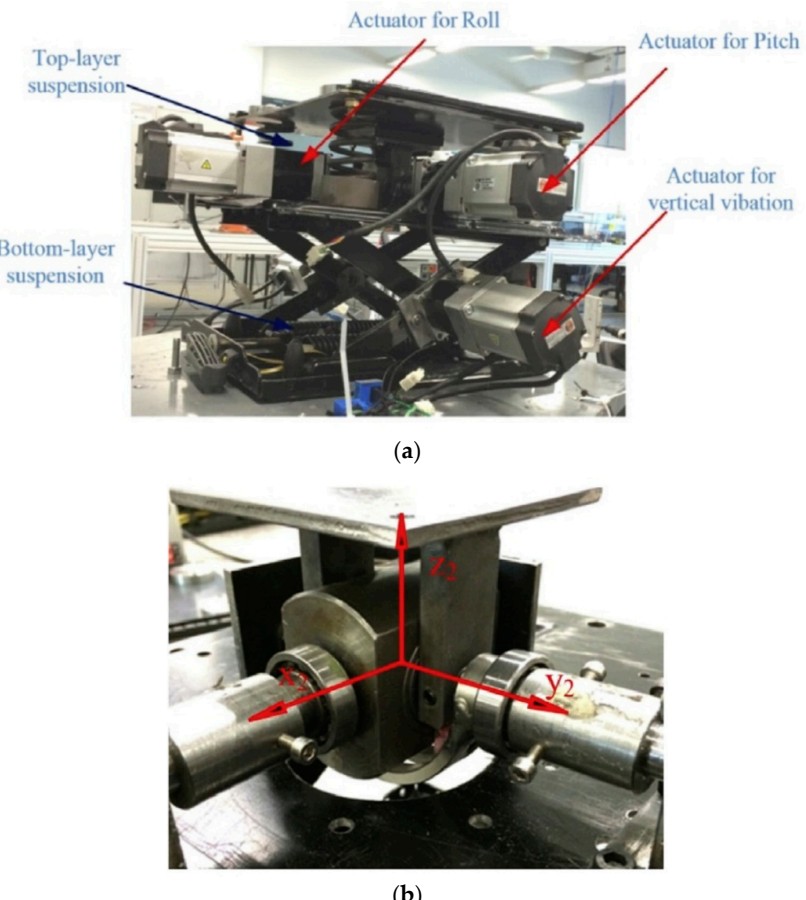

(a)

(b)

**Figure 15.** (**a**) The double-layer seat suspension prototype with multi-DOF vibration control mechanism (**b**) The universal joint [61].

In another project, with the same double-layer active seat suspension, an associated motion controller was designed by Ning et al. (2018) [62] to reduce the WBV of commercial vehicle drivers. The proposed active seat suspension can attenuate vibrations in 5-DOF, except the yaw vibration. The experimental results showed that when the seating system is fully controlled, the WBV can be reduced over 40%.

An active seating system vibration control (Figure 16) using an electromagnetic linear actuator and a related controller was designed by Gan et al. (2015) [63]. The system uses a combination of active and passive suspensions, where the passive suspension is primarily responsible for static load bearing, while

two electromagnetic active suspensions (XTA-3806) that can produce a peak force of 1116 N are placed at both ends of the seat to generate the force needed to reduce vibration. In the next step, an adaptive controller (Figure 17) based on the traditional filtered-X least mean square (FXLMS) algorithm combined with an on-line fast-block LMS identification method was developed. For this project, the conventional FXLMS method was used to deal with the time-varying and nonlinear nature of the system, and the narrowband feed-forward FXLMS algorithm was employed to reduce the narrowband vibration caused by mechanical equipment. Finally, an online identification using a Fast-block least-mean-square (FBLMS) controller was used to mitigate low-frequency periodic vibrations, and multiple two-weighted adaptive filters were used in the system to reduce multiple harmonics.

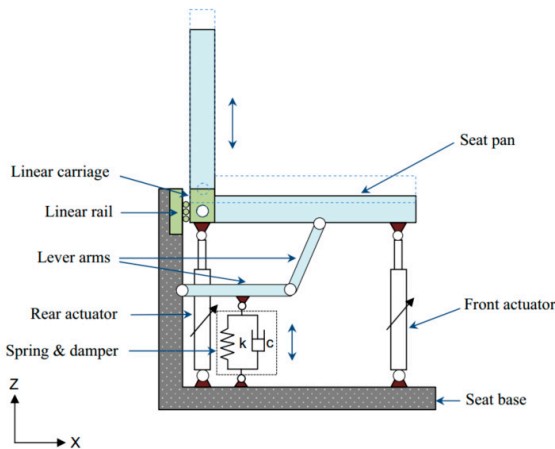

**Figure 16.** The model of the active seating system [63].

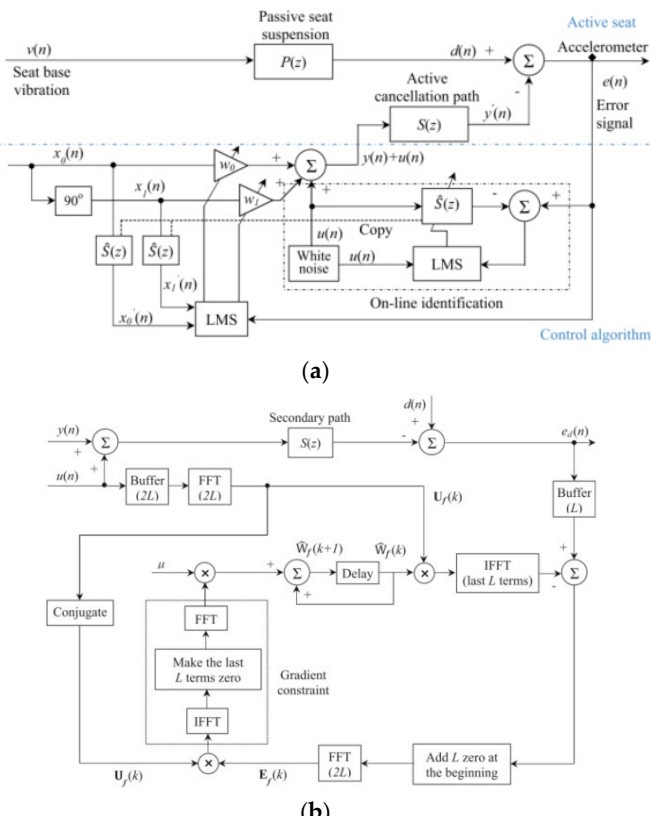

**Figure 17.** (**a**) The block diagrams of the filtered-X least mean square (FXLMS) controller (**b**) Fast-block least-mean-square FBLMS controller [63].

Frechin et al. (2004) [64] developed an active vibration control seating system and the related displacement compensation controller. The seat is mounted on a hemispherical motion base (Figure 18) to attenuate the 4-DOF vibration control through the active seat suspension. The acceleration on each axis is measured by a set of acceleration sensors mounted on the seat, and the displacement in the corresponding direction is calculated by the controller (Figure 19). The controller then sends a signal to drive the actuator for displacement compensation to eliminate vibration. The final experimental results proved that this active seating system could reduce low-frequency vibration (2–6 Hz) and improve the comfort of drivers [64].

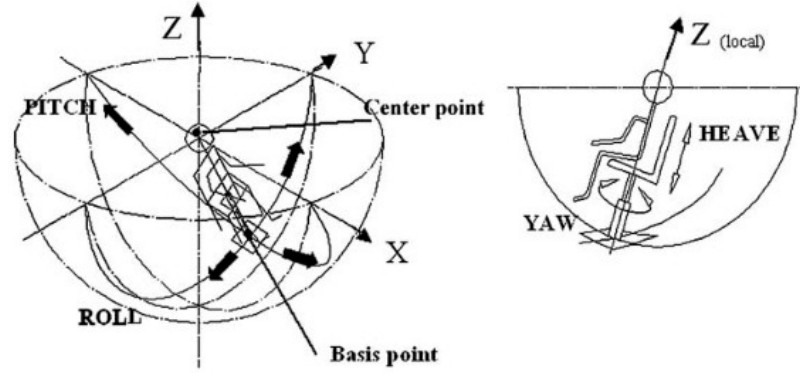

**Figure 18.** The semispherical motion base [64].

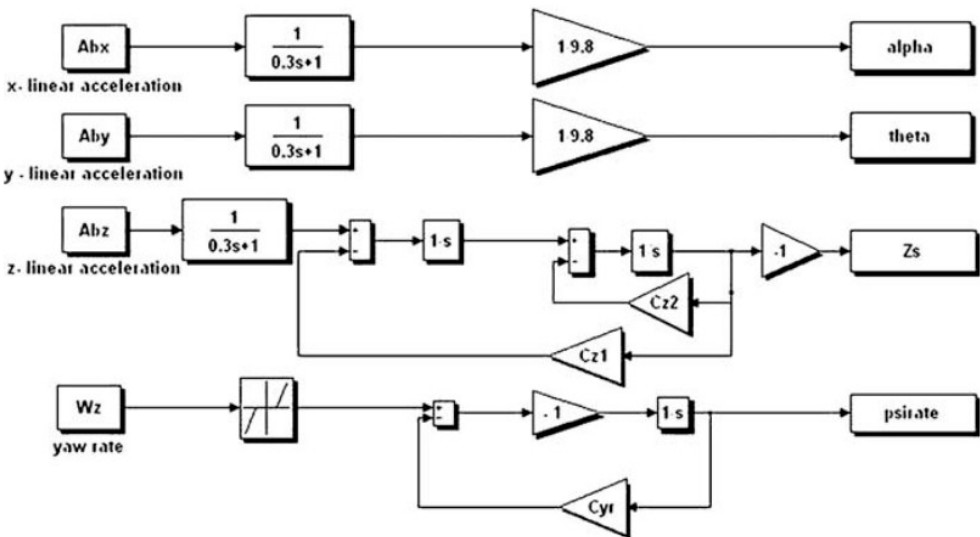

**Figure 19.** Control algorithm design [64].

The actuators used in the prototypes are summarized in Table 1.

**Table 1.** Summary of the actuators of active seat suspension systems.

| Author | Actuator and Driver | Degree-of-Freedom (DOF) Control | Max Output | Work Load | Pros and Cons |
|---|---|---|---|---|---|
| Stein (1997) | Pneumatic spring Proportional electropneumatic transducer | Vertical 1-DOF | | | Pros Simple structure The characteristics of the pneumatic spring itself help to reduce vibration Cons The pneumatic structure responds slowly Low control accuracy |

**Table 1.** *Cont.*

| Author | Actuator and Driver | Degree-of-Freedom (DOF) Control | Max Output | Work Load | Pros and Cons |
|---|---|---|---|---|---|
| Maciejewski et al. (2014) | Pneumatic spring | Vertical 1-DOF | | 55 kg 98 kg | Pros Common structure Traditional shock absorbers reduce energy consumption Cons Slow dynamic response Need pipeline and compressor |
| Maciejewski et al. (2010) | Pneumatic spring | Vertical 1-DOF | | 50 kg 80 kg 120 kg | Pros Common structure Traditional shock absorbers reduce energy consumption Cons Slow dynamic response Need pipeline and compressor |
| Maciejewski (2012) | Pneumatic spring | Vertical 1-DOF | Force = 400 N | 51 kg 102 kg | Pros Common structure Traditional shock absorbers reduce energy consumption Cons Slow dynamic response Need pipeline and compressor |
| Ning et al. (2016) | 400 W Panasonic servo motors ×2 (MSMJ042G1U)×2 | Vertical 1-DOF | Torque = 104 Nm | 80 kg | Pros Simple structure Responsive Easy to control Con Bulky |
| Ning et al. (2017) | 400 W Panasonic servo motors servo motor drivers (MBDKT2510CA1) | Vertical 1-DOF | Torque = 26 Nm | 55 kg 70 kg | Pros Simple structure Responsive Easy to control Con Bulky |
| Ning et al. (2017) | 400 W Panasonic servo motors servo motor drivers (MBDKT2510CA1) | Vertical 1-DOF | | | Pros Simple structure Responsive Easy to control Con Bulky |
| Ning et al. (2016) | 400 W Panasonic servo motors servo motor drivers (MBDKT2510CA1) | Vertical 1-DOF | Torque = 52 Nm | 55 kg | Pros Simple structure Responsive Easy to control Con Bulky |
| Ning et al. (2018) | 400 W Panasonic servo motors ×4 servo motor drivers (MBDKT2510CA1) ×4 | Vertical and roll 2-DOF | Torque = 52 Nm Force = 350 N | 80 kg | Pros Simple structure Responsive Easy to control Con Bulky |
| Stein and Ballo (1991) | Electrohydraulic | Vertical 1-DOF | | 75 kg | Pros High output power High control precision ConBulky |
| Gan et al. (2015) | Electromagnetic linear actuator (XTA-3086) ×2 | Vertical 1-DOF | Force = 1116 N | 55 kg | Pros Simple structure Responsive Easy to control Con High power consumption |

*2.2. Simulation*

In the study of active seat suspension of vehicles, in addition to using a prototype to verify the control algorithm, conducting simulations with professional software is also a common method to verify the feasibility of the controller design.

Du et al. (2013) [65] proposed a model combining automotive chassis suspension, active seat suspension, and the driver's body to analyze and achieve integrated vibration control. A static output feedback controller considering driver weight changes and actuator saturation was designed for an active seating system. The simulation results showed that this integrated control strategy for an active seat suspension system could improve comfort and robustness.

Different controllers are always compared to find out the most suitable control method for the active vibration control of the seating system. Wang and Kazmierski (2005) [66] developed a Very High Speed Integrated Circuit Hardware Description Language that Includes Analog and Mixed-Signal Extensions (VHDL-AMS) model for vibration control of a car seating system, including an active electromechanical actuator. In that study, five different controllers were compared and analyzed, and the optimal control (OC) algorithm was identified as providing the best results for vibration control. In another study, Al-Junaid et al. (2006) [67] proposed a seating system model (Figure 20) under active vibration control by the SystemC-A modelling technique. Four control algorithms were compared with the OC algorithm for vibration mitigation performance.

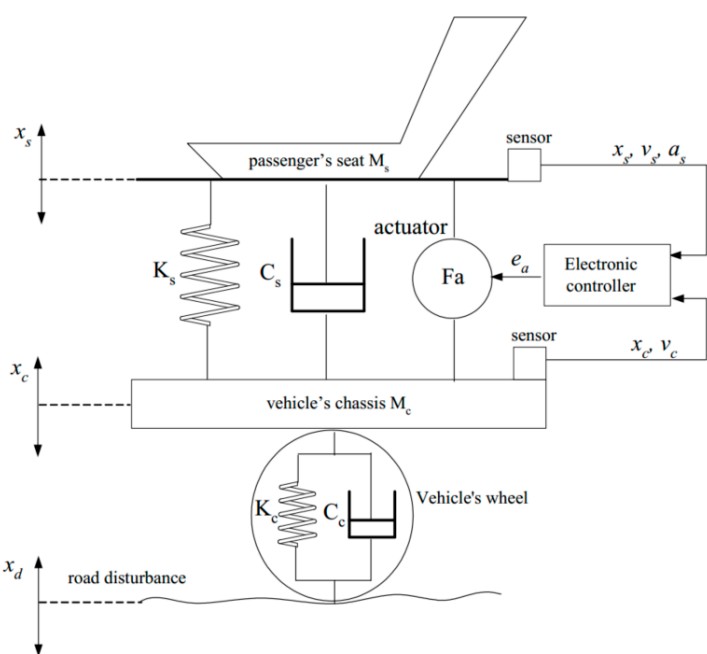

**Figure 20.** The schematic diagram of the model used in SystemC-A [67].

Wang et al. (2018) [68] proposed a parameter identification method to identify the system parameters of a 5-DOF discrete spring–mass–damper seating system model of a truck-based on truck field test data. The parameter identification method is based on trial and error to match the measured natural resonant frequencies and vibration acceleration amplitude at the selected frequencies with the simulated ones of the 5-DOF discrete spring–mass–damper seating system model. The disadvantage of the trial and error method is the inefficient parameter identification process, which requires much time and effort. The 5-DOF discrete spring–mass–damper seating system model can be used to simulate the vibration response of the human body. A sensitivity analysis was conducted using the Monte Carlo method based on the 5-DOF model. In that paper, primary and secondary PID controllers were applied to the seating system for active vibration control (Figure 21). The secondary PID controller produces a desired output control force signal according to the acceleration feedback signal of the mass oscillator.

The primary PID controller uses the relative displacement feedback signal and the signal generated from the secondary PID controller as input error signals. Then, the primary PID controller generates a final output control force signal to drive the actuator to provide control to mitigate vibrations of the seating system. The advantages of the PID control from that research are that it is simple and practical, but it also has the disadvantage of poor robustness.

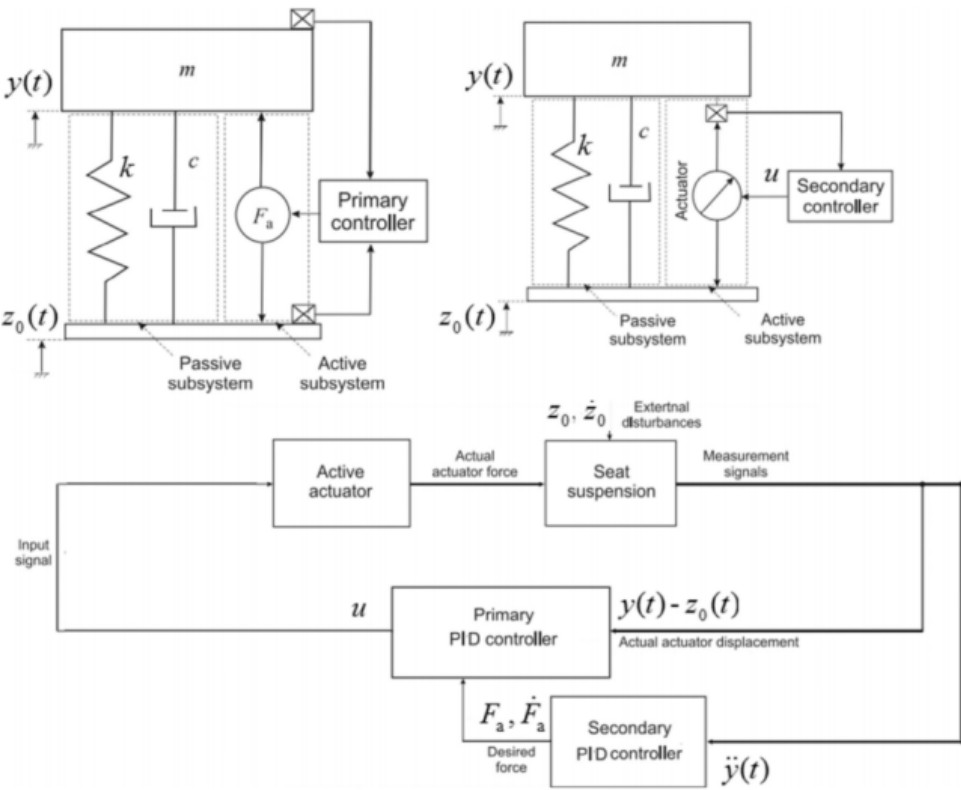

**Figure 21.** Block diagram of the proportional-integral-derivative (PID) control system [68].

Another PID control algorithm application for active vibration control of seating systems was proposed by Ali et al. (2019) [69]. In that study, a 13-DOF biodynamic model was designed for improving comfort and reducing the effects of low-frequency vibration on pregnant women and fetuses. For the controller design, the genetic algorithm was used to optimize the PID coefficients and the performance of the PID controller. In addition, the fuzzy PID (Figure 22) control algorithm was also used to design the controller to improve the robustness of the vibration control of the seating system. The performances of these two different PID controllers were compared through simulation with that of a seating system with a passive suspension. The results proved that the fuzzy PID controller in that study was the best for the low-frequency vibration control of the seating system.

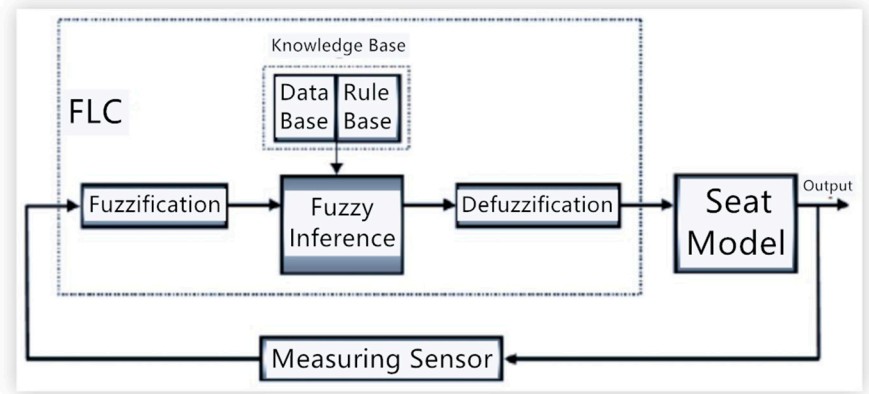

**Figure 22.** Fuzzy logic controller block [69].

A feed-forward adaptive controller (Figure 23) combined with the FXLMS algorithm and a feedback controller combined with the H-infinity algorithm was proposed by Wu and Chen (2004) [70] to reduce the small-amplitude vibration of a vehicle seating system. The researchers compared three different control methods and found that the FXLMS feed-forward adaptive controller alone could reduce the vibration amplitude of 4.8 dB at 10 Hz when it was turned on compared with the situation when the controller was turned off. The feedback controller with the H-infinity algorithm could reduce the vibration amplitude of 3.6 dB. Finally, the hybrid controller combined with the two control methods successfully reduced the vibration amplitude of 11 dB. The possible reason for this may have been that although the feed-forward adaptive FXLMS algorithm has the advantage of taking less computational time and easy implementation, it relies on the measurable reference signal. Once an unpredictable change of the vibration source occurs, this change greatly affects the convergence speed of the adaptive feed-forward controller algorithm. However, the feedback controller based on the H-infinity algorithm has good and robust performance. Therefore, the hybrid controller combining the advantages of the two algorithms should have the best and most robust performance for seat vibration control.

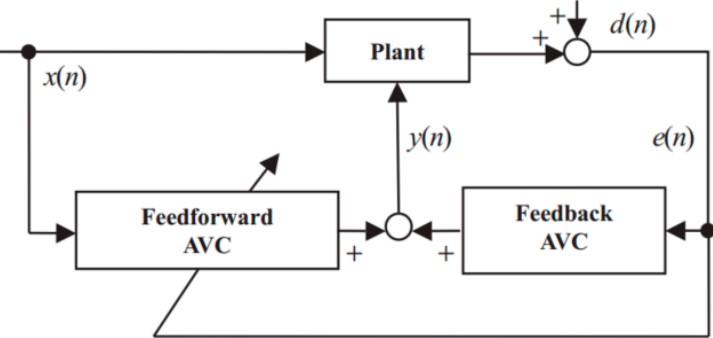

**Figure 23.** Schematic of the hybrid controller [70].

A feedback controller based on the H-infinity algorithm was proposed by Sun et al. (2011) [71] to reduce the vibration transmissibility of a seating system. A 3-DOF mass–spring biological model (Figure 24) was introduced to simulate the human body to improve the accuracy of the controller. Unlike the traditional H-infinity algorithm controller, the controller in this paper was optimized with a targeted frequency range by using the generalized Kalman–Yakubovich–Popov (KYP) lemma, which is mainly controlled in the sensitive frequency range of human body (4–8 Hz). According to the ISO 2631 standard, the D-class road profile roughness was simulated as the road surface excitation for the active suspension system. The simulation results showed that the system could significantly reduce vibration in the targeted range (4–8 Hz).

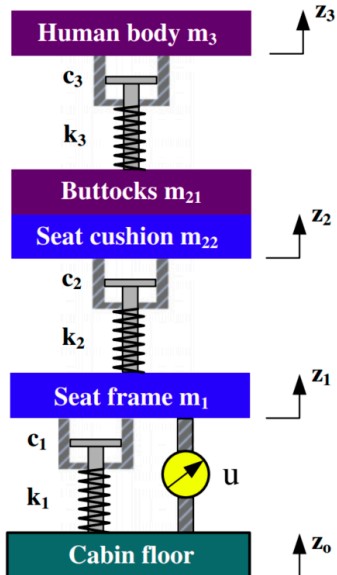

**Figure 24.** Three-DOF biodynamic model [71].

Du et al. (2012) [72] proposed an integrated control strategy that combines the driver body biodynamic model and the quarter car model to include active seat suspension and active vehicle suspension vibration controls to enhance ride comfort (Figure 25). In the controller design, the state feedback H-infinity controller of model simplification provides good robustness, considering friction as a feedback signal. According to the experiment results, the system can largely reduce the driver's head acceleration. It was shown that integrated active seat suspension and vehicle suspension controls could improve comfort compared with other separate controls.

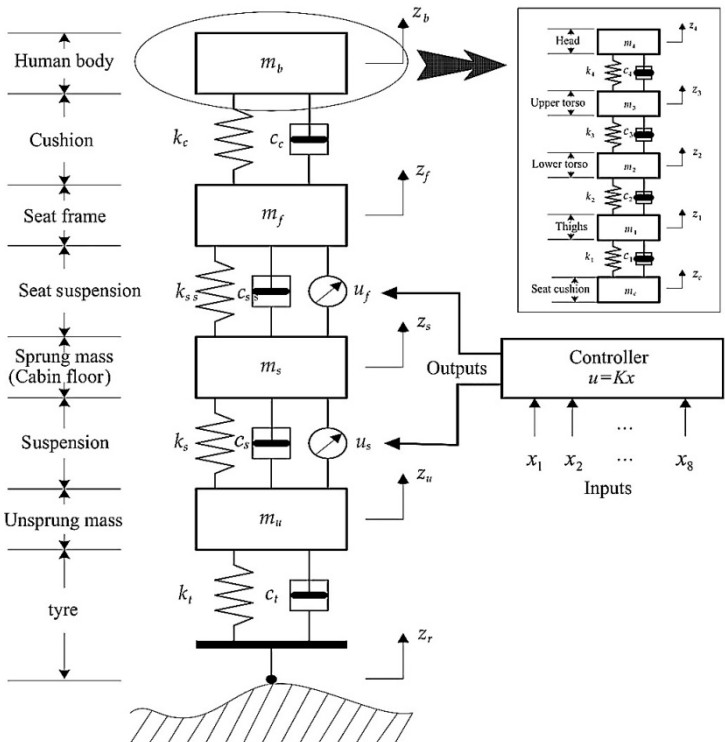

**Figure 25.** The model of the integral control strategy [72].

The different control system design details are summarized in Table 2.

**Table 2.** Summary of the control system designs.

| Author | Method | Target Frequency | Vibration Control Performance Criterion | Model | Performance | Pros and Cons |
|---|---|---|---|---|---|---|
| Wang et al. (2018) | PID control | 7.51 Hz | Displacement (m) | 5-DOF | The peaks of vibration amplitude are reduced to around $5 \times 10^{-10}$ m | Pros Simple Practical Con Poor robustness |
| Maciejewski et al. (2014) | Adaptive control Reverse model Proportional-Derivative (PD) control | 0.5–4 Hz | Seat Effective Amplitude Transmissibility (SEAT) (dimensionless) Transmissibility (Dimensionless) Acceleration (m/s$^2$) | 1-DOF | The vibrations at about 1.3 Hz are reduced by about 50% | Pros Adaptable Large range of load Con Signal delay |
| Maciejewski et al. (2010) | Robust control Triple feedback control | 0.5–4 Hz | SEAT (Dimensionless) Transmissibility (Dimensionless) | | The amplitude at resonance is reduced by about 50% | Pro Can work with a different mass load Con May cause chattering |
| Wu and Chen (2004) | FXLMS control H-infinity control | 10, 20, and 30 Hz | Accelerations in dB ref 1 m/s$^2$ | | The active seating system achieves 11 dB vibration attenuation at 10 Hz | Pro Good robustness Con May cause chattering |
| Ning et al. (2016) | H-infinity control with friction compensation | 1–4.5 Hz | Root-mean-square (RMS) acceleration (m/s$^2$) Frequency-weighted RMS (FW-RMS) acceleration (m/s$^2$) Vibration Dose Value (VDV) (m/s$^{1.75}$) SEAT (dimensionless) VDV ratio (dimensionless) | 1-DOF | Compared with a passive seating system, the RMS is reduced by 57%, the FW-RMS is reduced by 35.5%, the VDV value is reduced by 34.6%, the SEAT value is reduced 35.6%, and the VDV ratio is reduced by 34.6%. | Pro Good robustness Con Highly reliant on the accuracy of the model |
| Sun et al. (2011) | H-infinity control in the finite frequency domain | 4–8 Hz | Power Spectral Density (PSD) (m$^2$/s$^3$) Acceleration (m/s$^2$) | 3-DOF | | Pro Good robustness Con Highly reliant on the accuracy of the model |
| Gan et al. (2015) | FXLMS control FBLMS control | 4–12 Hz | dB ref m/s$^2$ | | For single-frequency cancellation, a 26 dB cancellation is achieved on the seat pan at the frequency of 6 Hz. For the multiple harmonic cancellation, the average level of vibration reduction is around 20 dB at 4, 6, 8, and 12 Hz. | Pro Adaptable Con Can be affected by noise or disturbance |

**Table 2.** *Cont.*

| Author | Method | Target Frequency | Vibration Control Performance Criterion | Model | Performance | Pros and Cons |
|---|---|---|---|---|---|---|
| Ning et al. (2016) | H-infinity control | 2–6 Hz | RMS acceleration (m/s$^2$) <br> FW-RMS acceleration (m/s$^2$) <br> VDV (m/s$^{1.75}$) <br> SEAT (Dimensionless) <br> VDV ratio (Dimensionless) | 2-DOF | Compared with a passive seating system, the RMS is reduced by 31.96%, the FW-RMS is reduced by 43.42%, the VDV value is reduced by 42.96%, the SEAT value is reduced 43.41%, and the VDV ratio is reduced by 42.68%. | Pro <br> Good robustness <br> Con <br> Highly reliant on the accuracy of the model |
| Ning et al. (2018) | Sliding mode control <br> H-infinity control | | RMS (m/s$^2$) <br> FW-RMS (m/s$^2$) <br> VDV (m/s$^{1.75}$) | 2-DOF | Compared with a passive seating system, for vertical acceleration, the RMS is reduced by 41.9%, the FW-RMS is reduced by 32.1%, and the VDV is reduced by 32.8%. For lateral acceleration cancellation, the RMS is reduced by 55.4%, the FW-RMS is reduced by 49.4%, and the VDV is reduced by 52.2%. | Pros <br> Adaptable <br> High robustness <br> Con <br> Highly reliant on the accuracy of the model |
| Ning et al. (2017) | Disturbance observer <br> Takagi–Sugeno fuzzy control | 2–4 Hz | Acceleration (m/s$^2$) <br> Transmissibility (Dimensionless) | 2-DOF | Compared with a well-tuned passive seating system, the active seating system can reduce the RMS by 45.5% and 49.5% with a mass load of 55 and 70 kg, respectively. | Pro <br> Effectively reduce the workload |
| Du et al. (2012) | H-infinity state feedback control | | Head acceleration (m/s$^2$) | 8-DOF | | Pro <br> Good robustness <br> Con <br> Highly reliant on the accuracy of the model |
| Ning et al. (2017) | Sliding mode control | 1.5 Hz | RMS acceleration (m/s$^2$) <br> FW-RMS acceleration (m/s$^2$) <br> VDV (m/s$^{1.75}$) <br> SEAT (Dimensionless) <br> VDV ratio (Dimensionless) | 2-DOF | Compared with a passive seating system, the RMS is reduced by 54.6%, the FW-RMS is reduced by 34.1%, the VDV value is reduced by 32.6%, the SEAT value is reduced 34.1%, and the VDV ratio is reduced by 32.6%. | Pros <br> Adaptable <br> High robustness <br> Con <br> Highly reliant on the accuracy of the model |

### 3. ANN Control

Unlike a model-based controller, the ANN controller is data-based. It can be used to identify complex nonlinear objects that are difficult to accurately model and can be used as a controller for adaptive control. Compared with traditional controllers, ANN controllers have many features. They have powerful nonlinear processing capabilities and are well suited for dealing with problems with a large number of input variables, as well as multivariable output. In addition, the ANN can learn unknown information autonomously. However, for active vibration control of a seating system, the ANN system has not been widely accepted.

An ANN controller (Figure 26) was proposed by Guclu and Gulez (2008) [73] to control a nonlinear vehicle model having 8 DOF. In this study, the active seat suspension system control was combined with an active vehicle suspension system control. The ANN controller could solve the nonlinear problem caused by the vehicle system under excitation disturbances. The ANN model was trained with errors between actual and expected output results by using backpropagation. The results demonstrated that the ANN controller performs well at controlling seating system vibration.

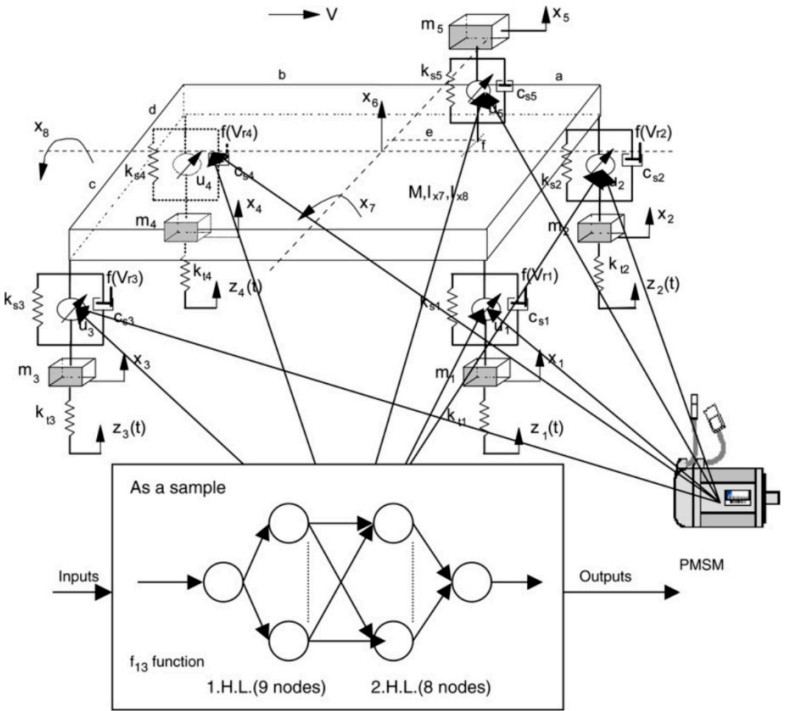

**Figure 26.** An integral controller based on artificial neural networks (ANNs) for active chassis suspension and active seat suspension system controls [73].

The control system for an active seat suspension combining an ANN module, the active force control (AFC) method, and a PID controller (Figure 27) was proposed by Gohari and Tahmasebi (2015) [74]. In the design of the ANN model, a hidden layer structure of 10 neural units was considered to have the best performance. Reverse learning was used to train the ANN model to estimate the mass of the seat–occupant system. In addition, the PID controller was designed to work with minimal disturbance and at low speed, and the AFC controller was used to improve the robustness and vibration control performance of the control system. In this controller, the system error was taken as the input, and the estimated mass was set as the output. The Levenberg–Marquardt algorithm was used to provide numerical solutions for nonlinear minimization. After comparison with the PID controller, the ANN control performed better than the other controllers.

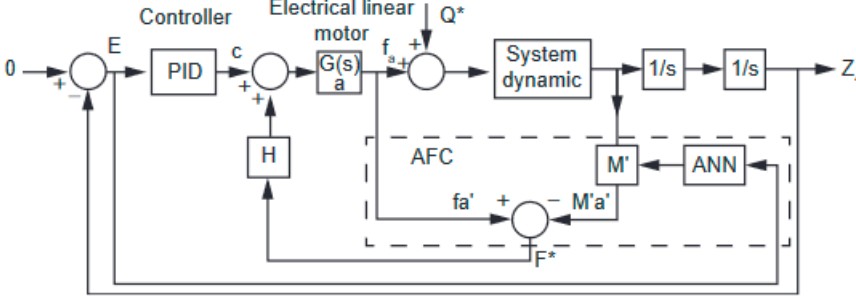

**Figure 27.** An intelligent controller via an ANN algorithm [74].

The ANN controllers mentioned here are summarized in Table 3.

**Table 3.** Summary of ANN controllers.

| Author | Number of Hidden Layers | Number of Nodes | Training Method | | Pros and Cons |
|---|---|---|---|---|---|
| Guclu and Gulez (2008) | 2 | First layer: 9 Second layer:10 | Back Propagation | Compared with an uncontrolled system, the maximum displacement of the passenger seat with NN control is reduced from $2.8 \times 10^{-3}$ to $0.2 \times 10^{-3}$ m. | Pro Good on multi-input and multioutput control Con Low robustness |
| Gohari and Tahmasebi (2015) | 1 | 10 | Back Propagation | | Pro Good on nonlinear problems Con Need large ideal training data |

## 4. Biodynamic Modeling

In the study of the active vibration control of a vehicle seating system, an accurate biodynamic model which aims to predict the dynamic response of a human body is necessary because it can provide more efficient control. In [68], a 5-DOF seat–occupant model (Figure 28), which can be used to simulate the dynamic behaviour of the human body, was proposed and the parameter sensitivity was determined based on the model; the relevant data were also used in the design of the active control system. In [71], a 2-DOF biodynamic model and a 2-DOF seat suspension system model were combined, and a 3-DOF seat–occupant model was established to describe the dynamic response of the human body in space. However, due to the difficulty of measuring actual human body properties and because an accurate biodynamic model may dramatically increase the amount of computation time, a simplified model has been used as a common method in research. In [59,60], the human body was replaced by a mass, and the amount of calculation time was reduced by ignoring the complex dynamic behavior of the human body itself.

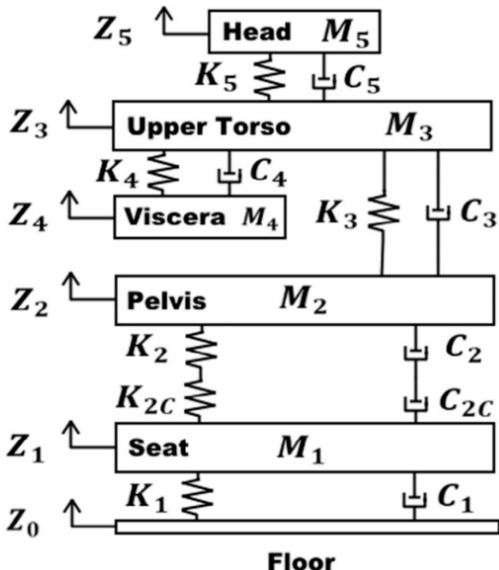

**Figure 28.** A 5-DOF seat–occupant model [68].

## 5. Our Contributions

Our group has been working on the active vibration control of a seating system since 2016. A 5-DOF seat–occupant biodynamic model was developed and applied in vibration control simulation research [68]. A new active vibration controller for a seating system that combines a traditional rotating and a scissor-like structure was designed and built to reduce low-frequency vibration and increase ride comfort [57]. The PID and H-infinity controllers were designed and applied to the active vibration control seating system in [57,68]. An analysis of the parameter sensitivity of a 5-DOF model-based Monte Carlo simulation was performed [68]. The measurement data and results recorded from four actual trucks in a field test were applied in the research to identify the 5-DOF system parameters of the human body seating system [68]. The relevant active vibration control of the seating system was simulated in the Simulink software [68].

*Identified Research Gaps, Research Questions, and New Directions*

In previous research, a number of innovations in terms of the active vibration control of a seating system have been proposed. Vibration control of a seating system has been well studied and analyzed in the laboratory. However, there are still some research areas that could be improved or gaps that need to be filled.

In terms of active vehicle suspension control, there are at least four active control algorithms that have been applied to the active vibration control of seating systems. However, the vibration control performance has not been crosswise compared to these various controllers. Such a project may allow researchers to identify which control algorithm is best for vibration control performance.

Due to the particularity of the human body, it is difficult to identify its model parameters in real-time. So, a real-time parameter identification method for human body biodynamic model parameters will help improve controller performance.

Compared with traditional controllers, ANN control reduces many of the modelling and calculation steps and simplifies controller design. However, ANN training requires a large amount of ideal data for back-propagation training. Therefore, determining how to use an unsupervised learning algorithm in place of a traditional supervised learning algorithm is an important future research direction.

An ANN control system has weak robustness due to sudden and unmanageable signals. Therefore, combining the ANN system with a traditional control system, using the control results of the traditional system to train the ANN controller, and enhancing the robustness of the system are also research areas worthy of consideration.

From the above-identified research gaps, the following research questions are raised:

1. What kind of ANN algorithm can be used to improve the PID control algorithm for active vibration control seating systems?
2. How can an unsupervised deeper learning algorithm be used to improve the performance of vibration cancellation in active seating systems?
3. How can an ANN algorithm be used to improve the robustness of a control system?

## 6. Conclusions

This article reviewed recent research on and developments in active vibration control in seat suspension systems. The advantages and disadvantages of each of the actuators and control algorithms were discussed. Examples of ANN control technology in the active vibration control of seating systems were also illustrated. Finally, this article identified the research gaps and new research directions which have not been covered by recent works, and research questions have been raised based on the identified gaps and research directions.

**Author Contributions:** Y.Z. analyzed the data; Y.Z. and X.W. contributed reagents/materials/analysis tools; Y.Z. and X.W. wrote the paper.

**Funding:** This research was funded by Australian Research Council grant number LP160100132.

**Conflicts of Interest:** The authors declare no conflict of interest.

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
