# Peer review of "A Review of Low-Frequency Active Vibration Control of Seat Suspension Systems"

_applsci, doi:10.3390/app9163326_

Round 1

Reviewer 1 Report

This review paper addresses a very good area and will be a worthy contribution to the literature. The structure and comparison table is good too. However, 35 reference is very low coverage of literature for a review paper. I would expect high tens or even 3 digit references for a good review paper. I would suggest the authors to expand more to have fuller coverage of the field, as well as wider vibration and control literature. The current manuscript is a good starting point, but a lot more work is needed to reach the threshold of a proper review paper.

Author Response

Reviewer 1

Question:

“This review paper addresses a very good area and will be a worthy contribution to the literature. The structure and comparison table are good too. However, 35 reference is very low coverage of literature for a review paper. I would expect high tens or even 3 digital references for a good review paper. I would suggest the authors to expand more to have fuller coverage of the field, as well as wider vibration and control literature. The current manuscript is a good starting point, but a lot more work is needed to reach the threshold of a proper review paper.”

Answer: This paper focuses on the active seating system vibration control algorithms using the machine learning including artificial neural network (ANN) control algorithms. Extra 41 reference papers have been added.

In Page 2, the following paragraph has been added:

“According to the study in [36], it is proved that the whole-body vibration (WBV) will affect the posture control of the human body and may cause health risk to the muscular system and the spine. In the study [37], it is shown that if the exposure of low-frequency vibrations produced by commercial vehicles is more than eight hours every day, conventional vehicle seating with passive suspension could not protect the driver's body from the effects of WBVs.”

In Page 5, “Passive vibration control of seating system can reduce vibration by using conventional springs and dampers components to be seat suspension system,” has been added.

In Page 6, “In [61], a negative stiffness seat suspension system combining with a pneumatic spring and the stiffness control mechanism was proposed, and the related control algorithm that affects device stiffness variation associated with the position and velocity data evaluation was designed.”

In Pages 6-7, “In terms of structure, in addition to the traditional shock absorber, the semi-active seat suspension is also designed in others different style. Bai, Jiang and Qian (2017) [62] designed an integrated semi-active seat suspension that includes a swing mechanism (Figure 3) that converts longitudinal and vertical motion into rotational motion and a torque-controlled rotary MR damper operated in pure shear mode to attenuate vertical and longitudinal vibrations. Bai, Jiang and Qian (2017) [63] proposed a new semi-active seating suspension based on the variable admittance (VA) concept and design a rotating VA device based on magneto-rheological damper. The random vibration test shows that the semi-active seating suspension has excellent performance of low-frequency vibration cancellation; the frequency-weighted root mean square acceleration of the seat is reduced by 43.6%, indicating that the ride comfort is greatly improved. Ning, Donghong et al. (2018) [64] developed a semi-active vibration control seating system based on an energy regeneration device with variable external resistance. The energy regeneration seat device includes a three-phase generator and a gear reducer mounted at the centre of the scissor-liked structure of the seat, and the vibrational energy is collected directly from the rotational motion of the scissor-liked structure. A H-infinite state feedback controller is designed for the semi-active vibration control seating system and the frequency-weighted root mean square (FW-RMS) acceleration is reduced by 22.84% compared to a passive vehicle suspension. At the same time, the generated RMS power is 1.21W.” has been added.

In Page 7, Figure 3 has been added.

In Pages 7-9, “Comparing with the controller design based on single traditional adaptive control algorithm, more and more controllers integrated multiple control algorithms to the semi-active vibration control seating system has been proposed to improve vibration attenuation performance. Phu, Shin and Choi (2015) [65] developed a new adaptive fuzzy controller combining H-infinite algorithm and sliding-mode control algorithm for a semi-active seating suspension with a magneto-rheological fluid damper. This controller features a fuzzy control method that does not require an accurate dynamic model, even in an uncertain environment of a dynamic system. Phu, An and Choi (2017) [66] designed a new adaptive hybrid controller integrating the H-infinity control algorithm, the sliding-mode control algorithm and the PID control algorithm to the vibration attenuation of the semi-active seating system. This controller features a combination between the Hurwitz constant matrix as components of the sliding surface and H-infinity algorithm with robust stability. In addition, a fuzzy logic module based on interval type-2 fuzzy logic system is established and model is characterized by online clustering and considering external interference. Phu, Choi and Choi (2017) [67] proposed a new hybrid controller combining a neural fuzzy control module, PI control module and sliding mode control module to control a semi-active seating suspension with a MR damper. The interval type-2 fuzzy model with online rules updating function is adopted, and a granular clustering method is used to find the data to the initial fuzzy set used to support the fuzzy model. Compared to conventional controllers, the proposed controller can provide better stability to vibration control performance. Nguyen, Nguyen and Choi (2015) [68] designed a novel neuro-fuzzy controller (NFC) for a semi-active seat suspension system with a magnetorheological damper. This adaptive neuro-fuzzy inference system (ANFIS) is based on an algorithm called B-ANFIS algorithm combined with a fuzzy logic system (FIS). Comparing with the Sky-hook control theory, the NFC has better performance on improve ride comfort of the vehicle. In addition, the ability to track trajectories and transient response characteristics of the NFC is superior compared with conventional Sky-hook controllers. Phu, Quoc Hung and Choi (2017) [69] proposed a new adaptive fuzzy controller based on the inversely fuzzified values related to the H-infinity control algorithm to the vibration control of the semi-active seating suspension system with a magnetorheological damper and the Riccati-like equation with fuzzified values to achieve to enhance robustness of the system.”

In Page 9, “ Experiment with prototype

The active seat suspension of the prototype in the research project uses the electromagnetic, hydraulic or air actuator to generate a corresponding compensation force to achieve the vibration cancellation on finite number of frequency, thereby reducing the vibration acceleration amplitude and improving the comfort of the seating system. According to [70], the utilization of active vibration control suspension on the vehicle has better vibration cancellation performance than the passive seating suspension. In the study of [71], comparative experiments have demonstrated that active and semi-active seating suspensions can improve the comfort by approximately 50% compared with the passive seating suspensions.” has been added.

In Page 10, “Stein (1997) [76] developed an active seat suspension using a pneumatic spring and a corresponding feedback controller. According to the results, the active seat suspension using vibration compensation can reduce the vibration amplitude by 10 dB, which is about 3 times. The vibration transmission rate is reduced by 30%-40% compared to conventional passive seat suspension.” has been added.

In Page 14, “Maciejewski, Krzyzynski and Meyer (2018) [74] develop a horizontal active seat suspension using pneumatic muscles (Figure 8) for horizontal vibration control. An original control system (Figure 9) combined a primary controller and an inverse model module is designed to provide control signal to the pneumatic muscles and a PD control module to be used to speed up the signal. According to the final results, the proposed active seat suspension has higher performance than passive seat suspension for vibration attenuation in the 1-10 Hz frequency range.”

In Page 14, Figures 8 & 9 have been added.

In Page 22, “In another project, with the same double layers active seat suspension, the associated motion controller was designed by Ning, D. et al. (2018) [73] for reducing the whole body vibration (WBV) of commercial vehicle drivers. The proposed active seating suspension can attenuate vibrations in 5-DOF, except the yaw vibration. The experimental results show that the when the seating system is fully controlled, the WBV can be reduced over 40%.” has been added.

In Page 23, “Frechin, Ariño and Fontaine (2004) developed an active vibration control seating system and the related displacement compensation controller. The seat is mounted on a hemispherical motion base (Figure 18) to attenuate 4-DOF vibration control through a set of actuator. The acceleration on each axis is measured by a set of acceleration sensors mounted on the seat and the displacement in the corresponding direction is calculated by the controller (Figure 19). The controller then sends a signal to drive the actuator for displacement compensation to eliminate vibration. The final experimental results prove that the active seating system can reduce low-frequency vibration (2 to 6 Hz) and improve the comfort of drivers [72].” has been added.

In Page 23, “

Simulation

In the study of active seat suspension of vehicles, in addition to using a prototype to verify the control algorithm, conducting simulations by professional software is also a common method to verify the feasibility of the controller design.

Du, Li and Zhang (2013) [75] proposed a model combining automotive chassis suspension, active seat suspension and the driver's body to analyze and achieve integrated vibration control. A static output feedback controller considering the driver weight changing and actuator saturation is designed for the active seating system. Simulation results show that this integrated control strategy of the active seat suspension system can improve comfort and robustness.”

In Page 55-58, the following reference papers have been added:

“Slota, GP, Granata, KP & Madigan, ML 2008, 'Effects of seated whole-body vibration on postural control of the trunk during unstable seated balance', Clinical Biomechanics, vol. 23, no. 4, pp. 381-386.

[36]   BURDORF, A & SWUSTE, P 1993, 'THE EFFECT OF SEAT SUSPENSION ON EXPOSURE TO WHOLE-BODY VIBRATION OF PROFESSIONAL DRIVERS*', Annals of Work Exposures and Health, vol. 37, no. 1, pp. 45-55.

[37]   Choi, S-B, Nam, M-H & Lee, B-K 2000, 'Vibration Control of a MR Seat Damper for Commercial Vehicles', Journal of Intelligent Material Systems and Structures, vol. 11, no. 12, pp. 936-944.

[38]   Han, YM, Nam, MH, Han, SS, Lee, HG & Choi, SB 2002, 'Vibration Control Evaluation of a Commercial Vehicle Featuring MR Seat Damper', Journal of Intelligent Material Systems and Structures, vol. 13, no. 9, pp. 575-579.

[39]   Song, X & Ahmadian, M 2004, 'Study of Semiactive Adaptive Control Algorithms with Magneto-Rheological Seat Suspension', paper presented to <https://doi.org/10.4271/2004-01-1648>.

[40]   Han, YM, Jung, JY, Choi, SB, Choi, YT & Wereley, NM 2006, 'Ride quality investigation of an electrorheological seat suspension to minimize human body vibrations', Proceedings of the Institution of Mechanical Engineers, Part D: Journal of Automobile Engineering, vol. 220, no. 2, pp. 139-150.

[41]   Zhao, Y, Zhao, L & Gao, H 2010, 'Vibration Control of Seat Suspension using H∞ Reliable Control', Journal of Vibration and Control, vol. 16, no. 12, pp. 1859-1879.

[42]   Zhao, Y, Sun, W & Gao, H 2010, 'Robust control synthesis for seat suspension systems with actuator saturation and time-varying input delay', Journal of Sound and Vibration, vol. 329, no. 21, pp. 4335-4353.

[43]   Gao, H, Zhao, Y & Sun, W 2010, 'Input-Delayed Control of Uncertain Seat Suspension Systems With Human-Body Model', IEEE Transactions on Control Systems Technology, vol. 18, no. 3, pp. 591-601.

[44]   Ning, D, Sun, S, Du, H, Li, W & Zhang, N 2018, 'Vibration control of an energy regenerative seat suspension with variable external resistance', Mechanical Systems and Signal Processing, vol. 106, pp. 94-113.

[45]   Yao, HJ, Fu, J, Yu, M & Peng, YX 2013, 'Semi-active control of seat suspension with MR damper', Journal of Physics: Conference Series, vol. 412, p. 012054.

[46]   Phu, DX, Choi, S-B, Lee, Y-S & Han, M-S 2015, 'Vibration control of a vehicle’s seat suspension featuring a magnetorheological damper based on a new adaptive fuzzy sliding-mode controller', Proceedings of the Institution of Mechanical Engineers, Part D: Journal of Automobile Engineering, vol. 230, no. 4, pp. 437-458.

[47]   Li, W, Zhang, X & Du, H 2012, 'Development and simulation evaluation of a magnetorheological elastomer isolator for seat vibration control', Journal of Intelligent Material Systems and Structures, vol. 23, no. 9, pp. 1041-1048.

[48]   Wu, X & Griffin, MJ 1997, 'A SEMI-ACTIVE CONTROL POLICY TO REDUCE THE OCCURRENCE AND SEVERITY OF END-STOP IMPACTS IN A SUSPENSION SEAT WITH AN ELECTRORHEOLOGICAL FLUID DAMPER', Journal of Sound and Vibration, vol. 203, no. 5, pp. 781-793.

[49]   Sathishkumar, P, Jancirani, J & John, D 2014, 'Reducing the seat vibration of vehicle by semi active force control technique', Journal of Mechanical Science and Technology, vol. 28, no. 2, pp. 473-479.

[50]   Metered, H & Šika, Z 'Vibration control of a semi-active seat suspension system using magnetorheological damper', 10-12 Sept. 2014, pp. 1-7.

[51]   Huseinbegovic, S & Tanovic, O 'Adjusting stiffness of air spring and damping of oil damper using fuzzy controller for vehicle seat vibration isolation', 27-28 March 2009, pp. 83-92.

[52]   Kawana, M & Shimogo, T 1998, 'Active Suspension of Truck Seat', Shock and Vibration, vol. 5, no. 1.

[53]   Choi, Y-T & Wereley, NM 2005, 'Mitigation of biodynamic response to vibratory and blast-induced shock loads using magnetorheological seat suspensions', Proceedings of the Institution of Mechanical Engineers, Part D: Journal of Automobile Engineering, vol. 219, no. 6, pp. 741-753.

[54]   Sun, SS, Ning, DH, Yang, J, Du, H, Zhang, SW & Li, WH 2016, 'A seat suspension with a rotary magnetorheological damper for heavy duty vehicles', Smart Materials and Structures, vol. 25, no. 10, p. 105032.

[55]   Park, C & Jeon, D 2002, 'Semiactive Vibration Control of a Smart Seat with an MR Fluid Damper Considering its Time Delay', Journal of Intelligent Material Systems and Structures, vol. 13, no. 7-8, pp. 521-524.

[56]   Qiang, Z & Yaxun, Y 'Improved single neuron PID control for heavy-duty vehicle magnetorheological seat suspension', 3-5 Sept. 2008, pp. 1-3.

[57]   Liu, P, Xia, X, Zhang, N, Ning, D & Zheng, M 2019, 'Torque response characteristics of a controllable electromagnetic damper for seat suspension vibration control', Mechanical Systems and Signal Processing, vol. 133, p. 106238.

[58]   McManus, SJ, St. Clair, KA, Boileau, PÉ, Boutin, J & Rakheja, S 2002, 'EVALUATION OF VIBRATION AND SHOCK ATTENUATION PERFORMANCE OF A SUSPENSION SEAT WITH A SEMI-ACTIVE MAGNETORHEOLOGICAL FLUID DAMPER', Journal of Sound and Vibration, vol. 253, no. 1, pp. 313-327.

[59]   Sireteanu, T, Stancioiu, D & Stammers, CW 2002, Use of magnetorheological fluid dampers in semi-active driver seat vibration control.

[60]   Lee, CM, Bogatchenkov, AH, Goverdovskiy, VN, Shynkarenko, YV & Temnikov, AI 2006, 'Position control of seat suspension with minimum stiffness', Journal of Sound and Vibration, vol. 292, no. 1, pp. 435-442.

[61]   Bai, X-X, Jiang, P & Qian, L-J 2017, 'Integrated semi-active seat suspension for both longitudinal and vertical vibration isolation', Journal of Intelligent Material Systems and Structures, vol. 28, no. 8, pp. 1036-1049.

[62]   Ning, D, Sun, S, Yu, J, Zheng, M, Du, H, Zhang, N & Li, W 2019, 'A rotary variable admittance device and its application in vehicle seat suspension vibration control', Journal of the Franklin Institute, vol.

[63]   Ning, D, Sun, S, Du, H, Li, W & Zhang, N 2018, 'Vibration control of an energy regenerative seat suspension with variable external resistance', Mechanical Systems and Signal Processing, vol. 106, pp. 94-113.

[64]   Phu, DX, Shin, DK & Choi, S-B 2015, 'Design of a new adaptive fuzzy controller and its application to vibration control of a vehicle seat installed with an MR damper', Smart Materials and Structures, vol. 24, no. 8, p. 085012.

[65]   Phu, DX, An, J-H & Choi, S-B 2017, 'A Novel Adaptive PID Controller with Application to Vibration Control of a Semi-Active Vehicle Seat Suspension', Applied Sciences, vol. 7, no. 10, p. 1055.

[66]   Phu, DX, Choi, S-M & Choi, S-B 2017, 'A new adaptive hybrid controller for vibration control of a vehicle seat suspension featuring MR damper', Journal of Vibration and Control, vol. 23, no. 20, pp. 3392-3413.

[67]   Nguyen, SD, Nguyen, QH & Choi, S-B 2015, 'A hybrid clustering based fuzzy structure for vibration control – Part 2: An application to semi-active vehicle seat-suspension system', Mechanical Systems and Signal Processing, vol. 56-57, pp. 288-301.

[68]   Phu, DX, Quoc Hung, N & Choi, S-B 2017, 'A novel adaptive controller featuring inversely fuzzified values with application to vibration control of magneto-rheological seat suspension system', Journal of Vibration and Control, vol. 24, no. 21, pp. 5000-5018.

[69]   Guclu, R 2005, 'Fuzzy Logic Control of Seat Vibrations of a Non-Linear Full Vehicle Model', Nonlinear Dynamics, vol. 40, no. 1, pp. 21-34.

[70]   Bouazara, M, Richard, MJ & Rakheja, S 2006, 'Safety and comfort analysis of a 3-D vehicle model with optimal non-linear active seat suspension', Journal of Terramechanics, vol. 43, no. 2, pp. 97-118.

[71]   Frechin, MM, Ariño, SB & Fontaine, J 2004, 'ACTISEAT: Active vehicle seat for acceleration compensation', Proceedings of the Institution of Mechanical Engineers, Part D: Journal of Automobile Engineering, vol. 218, no. 9, pp. 925-933.

[72]   Ning, D, Du, H, Sun, S, Li, W & Zhang, B 2018, 'An Innovative Two-Layer Multiple-DOF Seat Suspension for Vehicle Whole Body Vibration Control', IEEE/ASME Transactions on Mechatronics, vol. 23, no. 4, pp. 1787-1799.

[73]   Maciejewski, I, Krzyzynski, T & Meyer, H 2018, 'Modeling and vibration control of an active horizontal seat suspension with pneumatic muscles', Journal of Vibration and Control, vol. 24, no. 24, pp. 5938-5950.

[74]   Du, H, Li, W & Zhang, N 2013, 'Vibration Control of Vehicle Seat Integrating with Chassis Suspension and Driver Body Model', Advances in Structural Engineering, vol. 16, no. 1, pp. 1-9.

[75]   Stein, GJ 1997, 'A Driver’s Seat With Active Suspension of Electro-pneumatic Type', Journal of Vibration and Acoustics, vol. 119, no. 2, pp. 230-235.“.

Reviewer 2 Report

It is good to be published but minor modification is needed.

The paragraph construction is not appropriate. The authors are to organize the paragraphs by gathering the sentences with the same subject. The length of each paragraph should also be similar.

It seems that there are wrong spellings. The author should check and correct them. For example, "neuron network".

Author Response

Reviewer 2

Question:

“It is good to be published but minor modification is needed.

The paragraph construction is not appropriate. The authors are to organize the paragraphs by gathering the sentences with the same subject. The length of each paragraph should also be similar.

It seems that there are wrong spellings. The author should check and correct them. For example, "neuron network".”

Answer: A minor modification has been done. The paragraphs have been reorganized by gathering the sentences under the same subject. The length of each paragraph has been adjusted, although the length of each paragraph is not necessary to be same. All wrong spellings have been corrected. However, in [32], “'Neural network” was used rather than “neuron network”. Therefore, the spelling of “'Neural network” is thought to be correct.

Round 2

Reviewer 1 Report

Authors have extensively updated the paper and I am happy to recommend publication